# Microseismic Precursors of Coal Mine Water Inrush Characterized by Different Waveforms Manifest as Dry to Wet Fracturing

**DOI:** 10.3390/ijerph192114291

**Published:** 2022-11-01

**Authors:** Rui Yu, Jiawei Qian, Liang Liu, Huasheng Zha, Nan Li

**Affiliations:** 1School of Mines, China University of Mining and Technology, Xuzhou 221116, China; 2Wangjialing Coal Mine, China Coal Huajin Group Co., Ltd., Hejin 043300, China; 3School of Earth and Space Sciences, University of Science and Technology of China, Hefei 230026, China; 4College of Oceanography, Hohai University, Nanjing 210098, China; 5The Fifth Geological Brigade of Jiangxi Geological Bureau, Xinyu 338000, China; 6State Key Laboratory of Coal Resources and Safe Mining, China University of Mining and Technology, Xuzhou 221116, China

**Keywords:** floor water inrush, microseismic monitoring, hybrid-frequency waveforms, wet cracks

## Abstract

Microseismic monitoring systems have been widely installed to monitor potential water hazards in limestone of the coal floor. The temporal and spatial distribution of rock fracture-induced microseismic events can be used as early warning indicators of potential water inrush from the coal floor. We established a microseismic monitoring system in the working face of Wangjialing coal mine. Besides traditional fluid-independent rock fracture-induced microseismic waveforms, fluid-dependent hybrid-frequency microseismic waveforms also play important roles in determining the microseismic precursors of water inrush. Hybrid-frequency microseismic waveforms have a sharp P wave and no obvious S wave phase. We infer that the first high-frequency signal is caused by the brittleness of the rock in the floor under the influence of the water pressure. The second low-frequency signal is caused by the water oscillations in the fractures. These hybrid-frequency waveforms represent the development of fracturing. In addition, the lifting height of the complete aquiclude above the confined water is very limited, and the water inrush from the floor is often closely related to these hidden faults. Therefore, the activation signal of hidden faults in the working face of coal mining can be monitored to effectively warn about the water inrush from the coal seam floor caused by faults. By analyzing different microseismic events, the monitoring and early warning of water disaster in the coal mine floor can be improved. This will help in taking measures in advance within the mine to ensure personnel safety and to reduce property losses.

## 1. Introduction

At present, mineral resource development has entered the deep mining stage, and the threat of high-pressure confined water in the Ordovician limestone underlying the coal seam floor is increasingly becoming serious [1,2,3,4,5]. Under the combined influence of mining stress and groundwater pressure, the coal seam floor becomes damaged and then a water inrush accident occurs [6,7,8,9]. Groundwater inrush is a serious threat in coal mining [10]. For example, on 10 March 2017 and 25 May 2017, two water inrush accidents occurred at Pan Er coal mine [11]. On 10 September 2018, a water inrush accident occurred at Xiaoyun Coal Mine [12]. On 28 March 2010, a serious water inrush accident occurred at Wangjialing coal mine [6]. All of them caused great casualties and economic losses.

In the monitoring and early warning of coal seam water inrush, the key problem is determining that the floor is damaged and cracks have formed. Eventually, water penetrates into the seepage channel. From this point of view, it can be considered that the water inrush from the coal seam floor is the result of instability due to rock fracturing, such as micro-fracture initiation, development and penetration induced by stress disturbances in the mining process [13]. When a rock fracture is unstable, microseismic signals will be generated.

Microseismic monitoring technology can dynamically describe the fracture development. By determining the temporal and spatial distribution and intensity of the microseismicity, the spatial location, frequency and size of the micro-fracture can be determined. However, earthquake waveforms are complex [14]. In general, the signals generated by the failure of brittle rock are easily identified. The signals related to the fluid–rock interaction are out of the ordinary [15]. Cheng et al. (2018) found that the microseismic signals in a water inrush channel are different from other microseismic signals in a large model test [16]. Tary et al. (2014) analyzed some unconventional waveforms during hydraulic fracturing [17].

Long-period signals and very-long-period seismic signals (VLPs) are widely recorded in volcano monitoring [18]. Hybrid-frequency waveforms (EHWs) were reported as a new type of waveform in hydraulic fracturing-induced seismicity [19]. These waveforms also are widely reported in volcano monitoring and hydraulic fracturing [20,21,22,23]. 

We installed a microseismic monitoring system in the working face of the floor of Wangjialing coal mine #2 of China Coal Huajin Group Co., Ltd. Wangjialing coal mine, especially working face 12,313, suffers from water inrush disasters. Different microseismic waveforms corresponding to different stages of water inrush from the coal mine floor have been distinguished. In particular, the hybrid-frequency waveforms manifesting as dry to wet fracturing have been monitored. We hope to achieve real-time dynamic monitoring and early warning by paying attention to this hybrid-frequency signal.

## 2. Data

### 2.1. Overview of Work Area

As shown in Figure 1, Wangjialing coal mine is located in southwest Xiangning County, Shanxi Province, adjacent to Hejin County. According to the outcrop and borehole exposure in the mine field, the strata are Quaternary, Tertiary, Permian, Carboniferous and Ordovician in descending order [24]. The mining coal seam is #2 of the Shanxi Formation.

The average thickness of the coal seam in working face 12,313 of Wangjialing coal mine is 5.5 m. The coal seam floor is about 71.5 m away from the underlying Ordovician limestone. The water-resistant layer is mainly fine-grained sandstone, mudstone, sandy mudstone, siltstone, Taiyuan Formation limestone and thin coal seams.

Working face 12,313 of Wangjialing coal mine contains many hidden faults, so there is a risk of Ordovician limestone water disasters in the process of coal mining. In order to ensure the safe mining of working face 12,313, we built a microseismic monitoring system. By analyzing the focal parameters of the microseismic earthquakes and by comparing the characteristics of different waveforms, the development of a floor failure zone and the evolution a water inrush channel can be analyzed in advance.

### 2.2. Microseismic Monitoring System

Microseisms are similar to natural earthquakes, but the energy of microseisms is weak. The magnitude is usually less than zero, mainly corresponding to micro-fractures caused by stress disturbances in the rock [25]. A microseismic monitoring system is mainly affected by three factors: the ambient noise, the epicenter distance and the source energy [26]. These three aspects affect the ability to record, identify and locate microseisms. In the process of coal mining, the energy of a micro-fracture signal induced by mining disturbances in the rock under the floor is weak, and the released energy attenuates exponentially with distance [27]. Therefore, the energy requirement for ambient noise in the monitoring environment is very high. The voltage of the ambient noise should not be greater than 0.01 mV.

In this study, the KJ648 microseismic monitoring system is used for monitoring and early warning of floor water inrush in working face 12,313 of Wangjialing coal mine. It has a high sensitivity (200 V/m/s), a wide frequency band (0.1–1000 Hz) and low ambient noise. Weak micro-fracture signals can be collected, so we can analyze the development of a floor failure zone and the evolution of a water inrush channel in advance.

### 2.3. Microseismic Monitoring Network

As shown in Figure 2, 23 one-component microseismic monitoring stations of working face 12,313 in Wangjialing coal mine are distributed in the upper and lower troughs. The distance between adjacent microseismic monitoring stations is about 90 m, and working face 12,313 is enveloped in a space for all-round monitoring. The bottom of each microseismic monitoring station is embedded in the bedrock coupled with cement grouting, so as to meet the requirements of microseismic monitoring and early warning of floor water inrushes.

## 3. Normal Microseismic Signals

From 15 December 2020 to 15 January 2021, the microseismic monitoring system of working face 12,313 in Wangjialing coal mine monitored 772 microseismic events located in the floor. These events were detected by the standard short-term average/long-term average (STA/LTA) algorithm [28] and located by the grid search method [29]. 

Then, we calculated the moment magnitude and the energy of the microseismic events [30]. The energy was mainly concentrated between 0 and 100 J, accounting for 75%; between 100 and 500 J, accounting for 17.88%; between 500 and 1000 J, accounting for 2.72%; and between 1000 and 5000 J, accounting for 3.89%. Most of their moment magnitudes were below 0. 

### 3.1. Time Domain Analysis of Floor Failure

The microseismic monitoring data of working face 12,313 at Wangjialing coal mine were analyzed day by day. From 15 December 2020 to 15 January 2021, the number of daily microseismic events differed, indicating that the underlying rock mass of the coal seam floor was damaged by mining disturbances. However, during the period from 15 December 2020 to 18 December 2020, the number of daily microseismic events increased sharply (the number of microseismic events in the floor on December 16 was 92), with an increase of about three to four times, indicating that the degree of damage in the underlying rock mass of the coal seam floor by mining disturbances was increasing. A seismic wave usually consists of body waves and surface waves. The body waves comprise primary pressure waves (P wave) and secondary shear waves (S wave). As shown in Figure 3, the microseismic waveform has obvious P wave and S wave phases. Significantly, we only show the sensors with clear waveforms. Compared with the P wave, the amplitude of the S wave is larger and more obvious. As shown in Figure 4, the dominant frequency of the microseismic waveform is about 90–200 Hz, determined by fast Fourier transform (FFT). 

### 3.2. Spatial Domain Analysis of Floor Failure

According to the microseismic data from monitoring the coal floor of working face 12,313 in Wangjialing coal mine, the following characteristics are shown in Figure 5: Firstly, the microseismic events of the underlying rock mass of coal seam #2 in working face 12,313 of Wangjialing coal mine are mainly distributed within 326.8 m in front of the working face. The high-energy events are mainly distributed near the f313-1, f32 and CF2 faults in front of the working face, which may be due to the activation of micro-fractures in the fault area caused by mining disturbances. As the distance between the mining face becomes closer to the fault anomaly area, the influence of mining disturbances on this area further increases. The number and energy of induced microseismic events increase too. When the mining face is far away from the fault area, the microseismic events in the area near the fault gradually decrease, and the impact of mining disturbances is small. Secondly, microseismic events in the coal seam floor are mainly concentrated in two areas. The microseismic event in the left part is mainly due to rock fracture signals caused by mining disturbances. The influence range is 89.2 m along the strike and 278.3 m along the dip, which is distributed 67.5 m in front of the coal mining area. The microseismic events in the right part present regular bands, about 303.7 m long and 66.4 m wide, connecting three faults (the f313-1, f32 and CF2 faults), which are jointly affected by faults and mining. Thirdly, based on an analysis of the tangential advancing position of working face 12,313 and the aggregation degree of microseismic events, the microseismic events in the coal seam floor are mainly concentrated in the fine-grained sandstone, mudstone, sandy mudstone, siltstone, limestone of Taiyuan Formation, thin coal seam (coal seams #3, #7, #8 and #10), and other aquifers between the floor and the limestone of Ordovician Fengfeng Formation. Rock micro-fractures appeared in the mudstone layer about 64.3 m below the floor successively, as shown in Figure 5. They are close to the Ordovician limestone with relatively developed karst fissures (about 71.5 m away from the floor of coal seam #2).

During the mining process of working face 12,313 in Wangjialing coal mine, the degree of stress concentration and the energy in the rock mass continue to accumulate and increase. In the process of stress balance and energy release, the micro-fracture of the underlying rock mass under the coal seam floor continues to occur, as shown in Figure 5. Compared with the goaf, the rock mass under the coal seam floor in front of the mining face is more damaged and the microseismic events are relatively more concentrated.

## 4. Hybrid-Frequency Waveforms

As shown in Figure 6, at 3:07 am on 17 December 2021, we found a different signal. The microseismic waveform had a sharp P wave and no obvious S wave phase, but the waveform energy was a little bit higher than that of a normal signal. Through the analysis of the time–frequency characteristics, the signal had a high-frequency head following by a low-frequency tail, which were especially obvious at station 18 and station 19. The dominant frequency of the waveform head was high, about 80–200 Hz, which is the same as normal signals. 

This feature is similar to the mixed frequency (hybrid frequency) seismic signals observed in volcanic seismic monitoring. These events usually have a sharp P wave and no obvious S wave phase. The first part has a high frequency, followed by a low-frequency part [22,31]. A common explanation is that the first high-frequency signal is caused by the brittleness of the rock in the floor under the influence of the water pressure. The second low-frequency signal is caused by the water oscillation in the fractures. Depending on whether there is fluid in the crack during fracture, its waveform is different.

In our case, a micro-fracture in a rock can produce cracks. When the crack penetrates to form a channel, water flows into this channel, and this process generates the low-frequency tail of a microseismic waveform with a dominant frequency of about 20 Hz, lasting about 2.1 s, as shown in Figure 7. If no fluid enters the channel, there will only be normal microseismic signals. If the fluid enters the channel but a channel does not develop in the floor, water inrush will temporarily not occur in the roadway. However, the situation has already become dangerous. In Figure 8, two observed hybrid-frequency waveforms are located at 291.8 m in front of the working face and are located deep under the Ordovician limestone of the coal seam floor.

## 5. Discussion

In the process of mining, the floor failure zone and the confined water-conducting zone develop dynamically. Firstly, the damage in the floor failure zone can produce normal microseimic signals, manifesting as dry fracturing. Secondly, as soon as the damage in the floor failure zone and the confined water-conducting zone connect, which creates a water inrush channel, hybrid-frequency signals appear. Flowing water in the channel may form a low-frequency harmonic signal.

From January 1, 2021 to January 7, 2021, a total of eight fault activation signals occurred in working face 12,313 of Wangjialing coal mine. As shown in Figure 9, these microseismic waveforms had clear P and S wave phases, and the S wave was well-developed, which is most probably caused by a shear fracture. As shown in Figure 10, through a time–frequency analysis, the main frequency was determined to be about 20–60 Hz. As shown in Figure 11, these events are of higher energy than normal microseimic signals and hybrid-frequency microseimic signals.

Fault water inrush make up a large part in Chinese water inrush accidents [32]. The combined action of faults and underground water pressure fractures and activates the surrounding rock, which provide important conditions for the formation of a water inrush channel. The existence of faults destroys the integrity of the surrounding rock and changes the original stress field. Fault activation can further develop into a fault fracture zone and the primary fracture zone of the floor rock, which connects with the floor limestone aquifer to form a water inrush channel and to induce floor water inrush. Under the influence of mining and water pressure, the redistribution of rock stress and the change in the shape of a failure zone in the coal floor may induce fault activation. In turn, the redistribution of stress and the change in the failure zone affect the pore pressure and seepage characteristics of groundwater, as shown in Figure 12.

## 6. Conclusions

To ensure personnel safety and to reduce property losses, the monitoring and early warning of floor water inrush during coal mining is carried out using the microseismic method. Based on different kinds of waveforms, we can determine the developmental stages of a water inrush disaster. Taking working face 12,313 of Wangjialing coal mine as an example, the following conclusions are drawn:(1)In the process of coal mining, most of the micro-fractures in rocks under the floor have obvious P waves and S waves, which are normal microseismic signals.(2)We detected a mixed frequency signal with a sharp P wave initiation, consisting of high frequency, followed by low-frequency harmonic coda. We estimate that the occurrence of these events led to two processes: Firstly, the initial failure of brittle rock under high fluid pressure is the reason for the high frequency part of the observed signal. Secondly, the fluid flows into the pore space, resulting in the resonance of a newly formed crack and the low-frequency harmonic coda.(3)The hidden fault under the coal seam floor has the characteristics of a small development scale, a small drop and a strong concealment, which mainly affect the original lifting height and mining lifting height of confined water in the aquifer. The lifting height of the complete aquiclude above the confined water is very limited, and the water inrush from the floor is often closely related to these hidden faults. Therefore, the activation signal of hidden faults in a coal mining face can be monitored to effectively warn about water inrush from the coal seam floor caused by faults.

## Figures and Tables

**Figure 1 ijerph-19-14291-f001:**
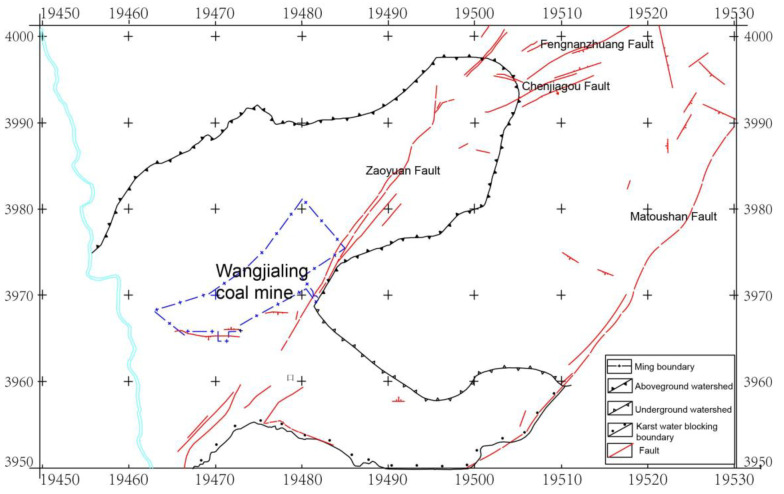
Location and structural geological map of Wangjialing coal mine. The blue line encircles Wangjialing coal mine.

**Figure 2 ijerph-19-14291-f002:**
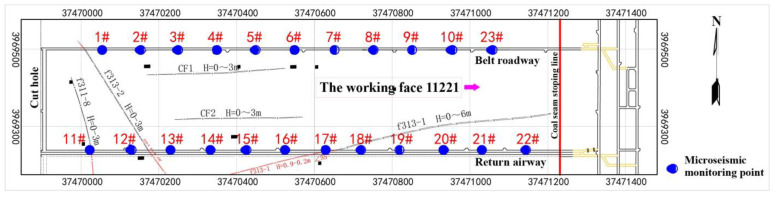
Microseismic monitoring network of working face 12,313 in Wangjialing coal mine.

**Figure 3 ijerph-19-14291-f003:**
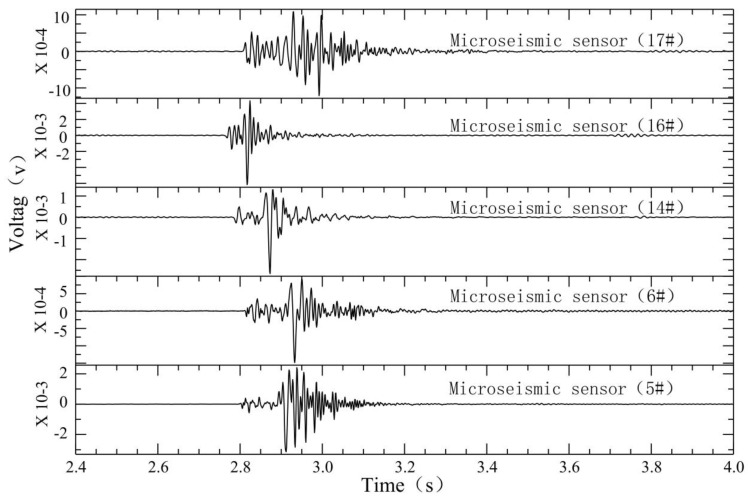
Normal microseismic signals induced by mining of working face 12,313 in Wangjialing coal mine.

**Figure 4 ijerph-19-14291-f004:**
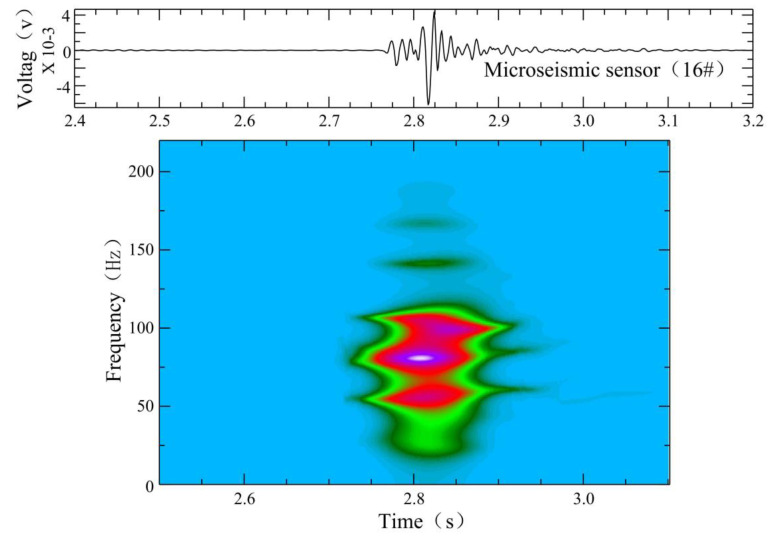
Fast Fourier transform (FFT) time–frequency analysis of a normal microseimic waveform induced by mining in working face 12,313 of Wangjialing coal mine. Sensor #16 was selected randomly. Warm color represents high decibel level, and cold color represents low decibel level.

**Figure 5 ijerph-19-14291-f005:**
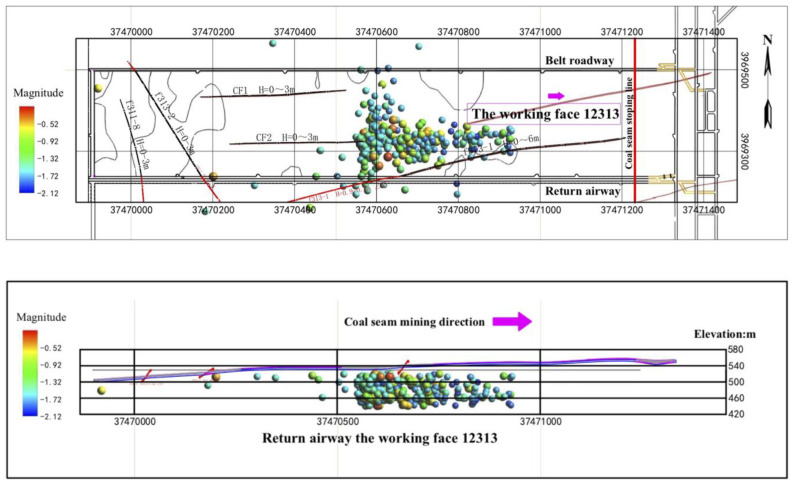
Spatial distribution of microseismic events (aerial view) of working face 12,313 of Wangjialing coal mine and spatial distribution of microseismic events along the trend section of working face 12,313 of Wangjialing coal mine. The color of the balls represents the moment magnitude.

**Figure 6 ijerph-19-14291-f006:**
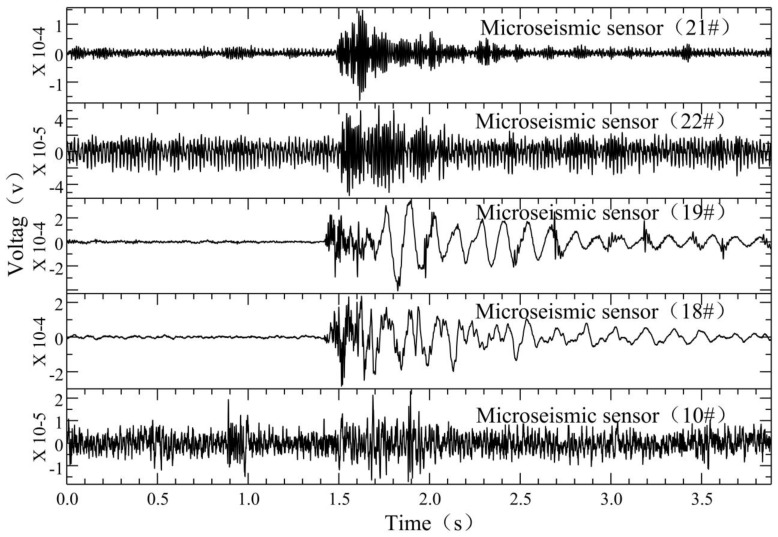
Hybrid-frequency waveforms of working face 12,313 of Wangjialing coal mine.

**Figure 7 ijerph-19-14291-f007:**
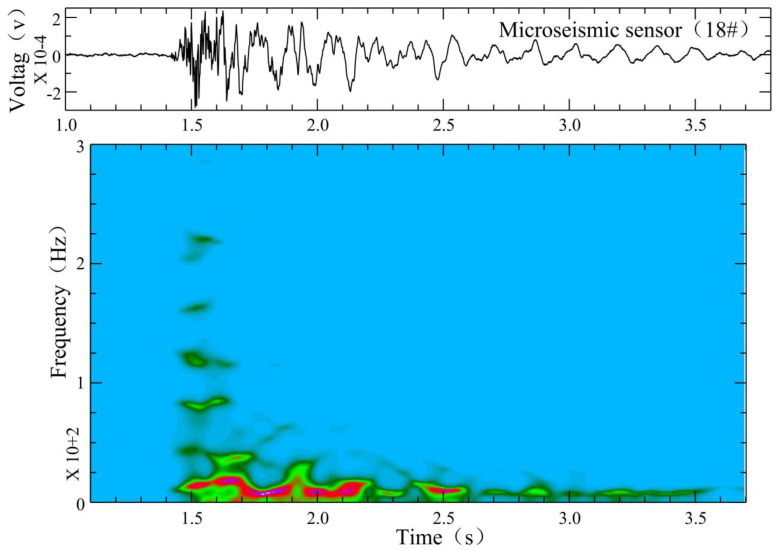
FFT time–frequency analysis of hybrid-frequency waveform of working face 12,313 of Wangjialing coal mine. Warm color represents high decibel level, and cold color represents low decibel level.

**Figure 8 ijerph-19-14291-f008:**
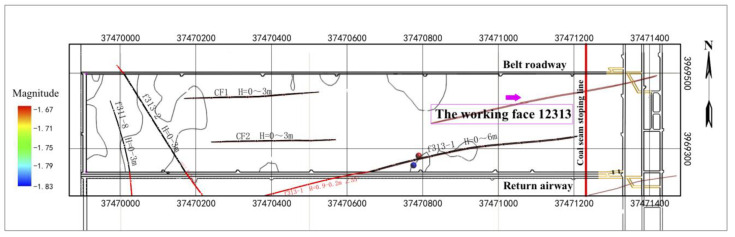
Spatial distribution of hybrid-frequency events of working face 12,313 of Wangjialing coal mine (aerial view).

**Figure 9 ijerph-19-14291-f009:**
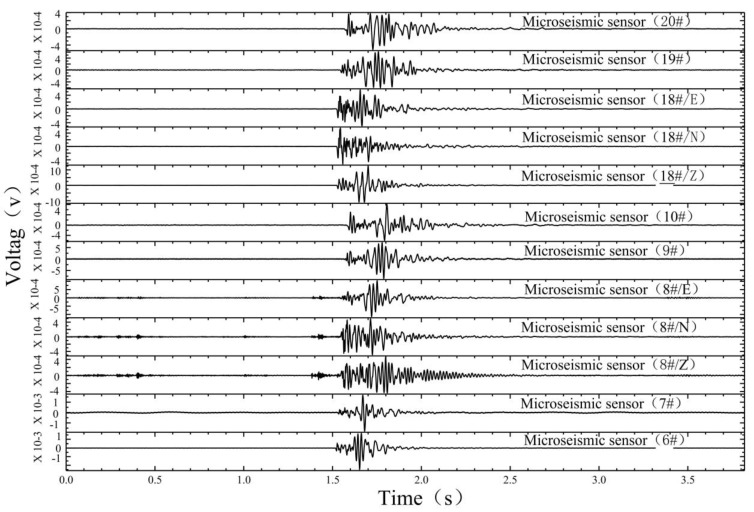
Fault activation signal waveforms of working face 12,313 of Wangjialing coal mine.

**Figure 10 ijerph-19-14291-f010:**
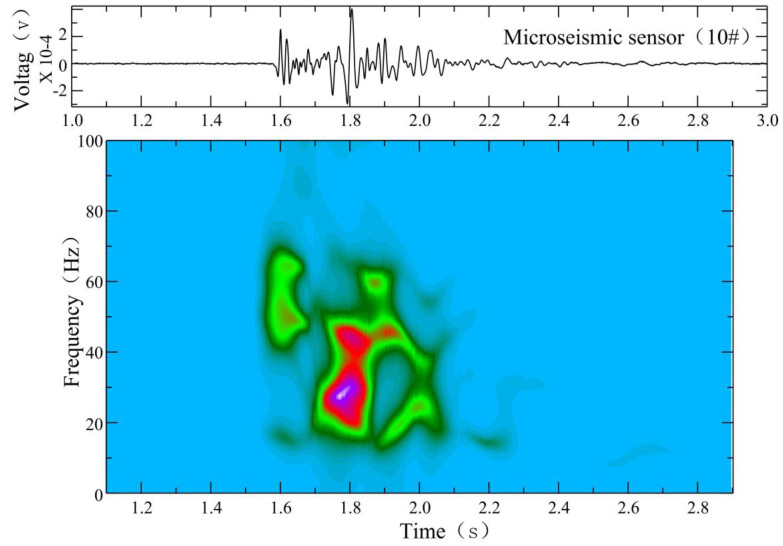
FFT time–frequency analysis of fault activation signal of working face 12,313 of Wangjialing coal mine. Warm color represents high decibel level, and cold color represents low decibel level.

**Figure 11 ijerph-19-14291-f011:**
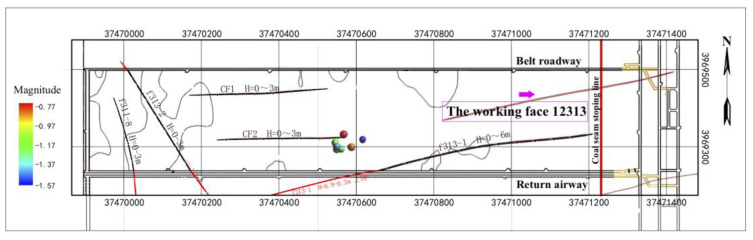
Spatial distribution of fault activation microseismic events (aerial view) of working face 12,313 of Wangjialing coal mine.

**Figure 12 ijerph-19-14291-f012:**
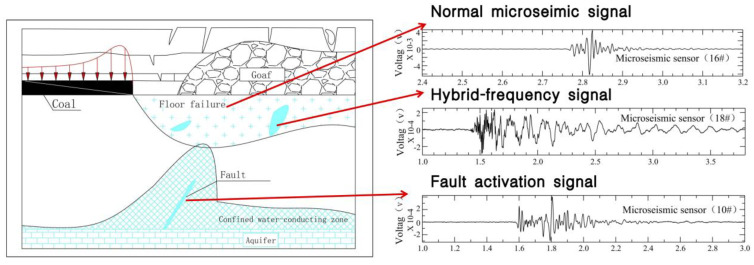
Water inrush model of a hidden fault under the coal seam floor and its link to the change in waveforms.

## Data Availability

The datasets analyzed or generated during the study can be obtained from the corresponding author.

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
