# Peer review of "Microseismic Precursors of Coal Mine Water Inrush Characterized by Different Waveforms Manifest as Dry to Wet Fracturing"

_ijerph, 2022, doi:10.3390/ijerph192114291_

Round 1

Reviewer 1 Report

In the text, "12313 working face of Wangjialing coal mine" suddenly appears - is it some specific working face? What was the basis for its selection? Is it the worst case from some particular point of view selected?

In the text, the authors present the results of seismic measurements about which I have no doubts. However, what is missing, in my opinion, in the discussion section of the results or the summary, is the potential of such measurements to determine the prediction of the formation of possible damage that may endanger the personnel and workers. Based on the FFT spectrum or recognition of the relevant P- and S-waves, are the authors able to determine the projected time of safe mining of a particular section of the mine deposit?

Reviewer 2 Report

1.The innovation is not clearly shown in the introduction.

2. The P wave and S wave need to be further explained in the paper.

3. In line 211, the hybrid-frequency signals will appear when the the existence of water inrush channel, please explain the specific reasons in detail. In the part 4, whether the hybrid-frequency signals are caused by water pressure is not verified, because it is not sure that water inrush occurred in the mine.

4. As for Fig.12, it is suggested that the water inrush model of hidden fault under coal seam floor should link with the change of wave.

5 In Fig.3, please explain why sensor 5#,sensor 6#, sensor 14#, sensor 16# and sensor 17# were selected. Similarly, are the sensors selected at randomit in the Fig.4, Fig.6, and Fig.9?

6. The conclusion of this article needs further modification, for example, in line 248, “We believe that  , it just could be treated as deduction.
